# Reaching silicon-based NEMS performances with 3D printed nanomechanical resonators

Stefano Stassi [1✉], Ido Cooperstein[2], Mauro Tortello [1], Candido Fabrizio Pirri [1], Shlomo Magdassi[2✉] & Carlo Ricciardi [1]

The extreme miniaturization in NEMS resonators offers the possibility to reach an unprecedented resolution in high-performance mass sensing. These very low limits of detection are related to the combination of two factors: a small resonator mass and a high quality factor. The main drawback of NEMS is represented by the highly complex, multi-steps, and expensive fabrication processes. Several alternatives fabrication processes have been exploited, but they are still limited to MEMS range and very low-quality factor. Here we report the fabrication of rigid NEMS resonators with high-quality factors by a 3D printing approach. After a thermal step, we reach complex geometry printed devices composed of ceramic structures with high Young's modulus and low damping showing performances in line with silicon-based NEMS resonators ones. We demonstrate the possibility of rapid fabrication of NEMS devices that present an effective alternative to semiconducting resonators as highly sensitive mass and force sensors.

[1] Department of Applied Science and Technology, Politecnico di Torino, Corso Duca Degli Abruzzi, 24, 10129 Torino, Italy. [2] Casali Center for Applied Chemistry, Institute of Chemistry, The Hebrew University of Jerusalem, Jerusalem 91904, Israel. ✉email: stefano.stassi@polito.it; magdassi@mail.huji.ac.il

 1

The continuous need for an increase in device performances and sensitivity brought to the shrinking of micro-electro-mechanical systems (MEMS) dimensions to the nanometric range and the development of nano-electro-mechanical systems (NEMS)[1–3]. The extreme device miniaturization in NEMS resonators offers the possibility to reach an unprecedented level of resolution in high-performance mass sensing[4,5] and force detection[6,7], as well as opening their implementation in quantum physics regime[8–10]. The very low limits of detection in mass and force spectroscopy reached by nanomechanical resonators are related to the combination of two factors: a small resonator mass and high-quality factor $Q$. A tiny device mass is needed such that small perturbations induce large resonance frequency variations. High-quality factor $Q$ means longer retention of coherent energy in the resonance mode which implies smaller frequency fluctuation that could mask the effect of the perturbations. Reaching such sensibility improvement, with low mass and high $Q$ devices, needs high complexity, multi-step, and expensive fabrication techniques that in some cases bring also to low fabrication yield, like for bottom-up devices such as CNT or graphene nanoresonators[11]. Several alternatives to standard semiconductor fabrication processes, like screen printing[12], hot embossing[13], microinjection molding[14], solvent casting[15], nanoimprinting[16], microfluidic approach[17], and 3D printing[18–21] have been exploited. Interesting devices have been realized with fast and low-cost techniques, often provided of intrinsic sensing functionality, due to the wide spectrum of fabrication materials investigated, compared to inert silicon-based materials that require a responsive coating to be implemented for sensing applications.

Another alternative technique to fabricate nanoresonators is two-photon printing (TPP) lithography[22–24]. Based on multiphoton absorption, the polymerization occurs only at the focal point of an ultrafast laser (780 nm), leading to selective submicron size voxel curing within a droplet, hence providing the ability to write sub-micrometric structures[25]. In contrast to better resolutions of single exposure photolithography techniques, such as photolithography and electron beam lithography, the TPP technique enables achieving complex 3D structures without the need for multi fabrication steps. Another advantage is the ability to integrate structures made of different materials in the same substrate, by changing the printing resin within the droplet[26,27]. One main disadvantage of TPP technology that prevented its adaptation in the industry compared to the traditional 2D printing techniques, is the slow printing process and the difficulty in making production at an industrial scale. However, due to the unique structures that can be fabricated by the TPP technology the industrial interest in this field is growing[28].

However, all these techniques have not been able to reach the sensing performances of silicon-based NEMS and bottom-up nanoresonators. All the resonators fabricated with alternative techniques are still limited to the MEMS range and do not reach nanometric dimensions, which means a large mass. In addition, the majority of the reported devices are composed of polymeric materials with low Young's modulus (in the range of 100 s MPa to few GPa)[29] and elevated loss factor, resulting in a very low-quality factor[30].

Here we report the fabrication of rigid NEMS resonators with high-quality factors by a 3D printing approach. The devices are printed by a two-photon polymerization technique to reach nanometric resolution with our recently developed liquid ink composed of metal salts and photopolymerizable groups[31]. The ink enables at first step a spatial photopolymerization, followed by a thermal step to remove the organic content, and to achieve the densification of the metal precursors, resulting in complex geometry devices composed of rigid ceramic structures with high

Young's modulus and low damping. The 3D printed NEMS resonators show quality factors up to 15,000 and mass sensitivity of 450 zg which are in line with the performances of silicon-based NEMS resonators. We experimentally demonstrate the possibility of rapid fabrication of NEMS devices by an easy technique that presents an effective alternative to semiconducting resonators as highly sensitive mass and force sensors.

## Results

**State of the art of MEMS and NEMS.** Following a detailed literature analysis, we looked at the relation between the quality factor (i.e., resonator coherence) and the mass of the device, the key factors which affect resonator sensitivities, and evaluated if our 3D printed devices can reach the standard silicon-based NEMS performances. Figure 1 presents the values of quality factors measured at room temperature as a function of device masses of a large set of resonators analyzed from the literature. In order to conduct as much as possible a comprehensive study, we analyzed more than 40 devices spanning over 16 orders of magnitude of device masses, divided into four different categories (detail of the analysis method and data references are reported in the Supplementary Information). Three categories are somehow related to standard lithographic technology: bottom-up NEMS (graphene, nanotube, and nanowires devices)[32–40], top-down NEMS[4,41–52], and MEMS resonators[53–57]. The fourth is represented by devices fabricated with the alternative techniques described previously[12–21]. For this analysis, we considered resonators of different geometries including membrane, clamped–clamped beams, and clamped free-beams (i.e., bridges and cantilever, respectively). As shown in Fig. 1, despite the different geometries, materials, and dimensions, a clear unique growing trend of the quality factor was found for the first three categories (proportional to resonator mass as $m^{1/3}$)[2]. While devices fabricated with alternative technologies show quality factors around two orders of magnitude lower concerning this trend. For this analysis, we do not consider highly stressed silicon

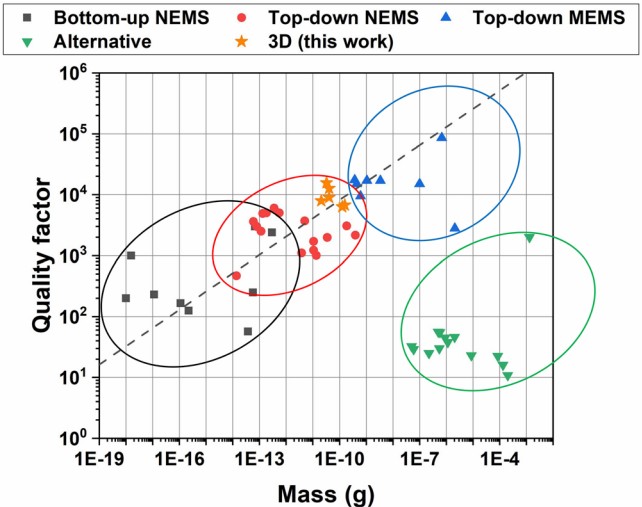

**Fig. 1 Quality factors of mechanical resonators from literature as a function of the fabrication method and device mass.** The quality factors of mechanical resonators at room temperature are extracted from the literature (references are reported in Supplementary Figs. S1, S2, and S3, and Supplementary Note 1) and divided regarding the fabrication method. Two of our devices with the best performances in terms of high $Q$ factor value and low device mass for each printed resonator structure (cantilever, bridge, and membrane) are reported as star points. The trend of $Q \propto m^{1/3}$ is reported as a dashed line.

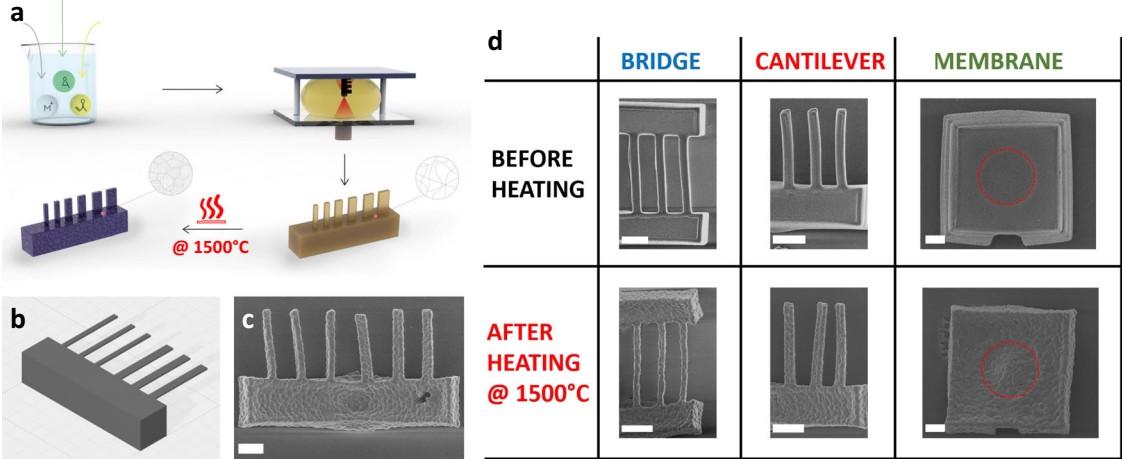

**Fig. 2 Scheme of the fabrication process and images of the printed NEMS devices. a** Scheme of the fabrication process starting from the precursor solution preparation, followed by TPP printing to photopolymerize locally the solution, structure washing, and final heating step to remove organic content and achieve densification and crystallization. **b** CAD scheme and **c** SEM image of a chip composed of six cantilevers of two different widths. **d** Images of the bridge, cantilever, and membrane devices before and after the heating step at 1500 °C. All the scale bars correspond to 10 μm.

nitrate devices and soft-clamped resonators based on dissipation dilution[58,59]. These approaches allow to obtain NEMS with a much higher quality factor of the reported trend (i.e., new record of $Q = 8 \times 10^8$) but need very complex fabrication techniques and large resonator size.

Thanks to our 3D printing approach, which is based on the conversion of soft hybrid structure into a rigid constitutive material with high-quality factor and low loss factor, we can surpass the performances of common devices which are fabricated with alternative techniques (stacked on MEMS size) and reach the trend of standard semiconductor-based NEMS, both in $Q$ values and device dimensions.

**Device fabrication**. Our devices are fabricated by printing a precursor solution ink with TPP technique, followed by an additional heating step at elevate temperatures to transform the structure from hybrid to rigid crystalline material (scheme in Fig. 2a, details in the "Methods" section)[31]. To prepare the precursor solution, metal chloride salts are first dissolved in an aqueous solution containing propylene glycol and acrylic acid. Upon addition of propylene oxide, the pH of the solution increased and initiates condensation between the metal ions to create metal oxide oligomers[60]. Due to the presence of the acrylic acid, a coordinative bond is formed between the metal ion and the acrylic acid, thus enabling a photopolymerization reaction by using suitable photoinitiators. After printing and washing the structures, the printed objects are heated to 1500 °C, first to remove the organic content, then to eliminate the pores, to achieve dense polycrystalline structures. Nd:YAG material is used for its high elastic modulus compared to the standard silicon-based NEMS[61], and to demonstrate the use of a material with intrinsic properties such as gain medium. To demonstrate the feasibility of rapid prototyping NEMS with performances comparable to their silicon-based counterparts, we printed NEMS resonators with the three most common designs: clamped–clamped beams (bridges), single-clamped beams (cantilevers), and circular membranes (Fig. 2b, c and d). We printed resonators of different dimensions, with lengths ranging from 20 to 50 μm, width from 2 to 5 μm, and thickness between 250 and 2000 nm. The dimensional control depends mainly on the printing parameters and the shrinkage of the printed object during the thermal process after printing. The TPP process enables printing objects having features as small as 100 nm[62]. In

our study, we start from a solution, obtain a hybrid object, followed by conversion of the hybrid structure (organic–inorganic) into an inorganic, dense crystalline structure. These processes lead to a significant shrinkage, and therefore it is theoretically possible to go down to features in the range of tens of nanometers. After the thermal post-printing process, the printed resonators are composed of only inorganic polycrystalline Nd:YAG (as reported by EDX spectrum before and after thermal step, as shown in Supplementary Fig. S4) without any organic materials. As a result of solvent evaporation, burning of the organic material, and crystallization to the dense crystal structure, the material sintering is accompanied by a dimensions reduction. To compare the actual dimensions with the computer design file, we calculated the ratio between the measured dimensions of printed structures after the thermal treatment process and the theoretical ones (used for the design). The size measurements were made by SEM imaging, and the analysis was computed over more than 200 resonators. The Gaussian fit reports a mean value of 68.7% of device isotopically shrinkage with a standard deviation of 5.3% (Supplementary Fig. S5). Although the size reduction can help to achieve very small features, it could result in deformation of the final device geometries, especially for the circular membrane which is the most complicated to fabricate due to stress-induced during the thermal process (image of a device broken by thermal stress in Supplementary Fig. S6). However, as it was presented in other publications, the deformation can be suppressed by printing the structures on guiding lines or domes[63,64]. Furthermore, after the thermal post-printing, the surface becomes rough due to the crystallization of the structure (as seen in Supplementary Fig. S6). To achieve a smoother surface, it is theoretically possible to gain smaller size grains by changing the heating conditions[65,66], selectively etch the YAG crystals with hot phosphoric acid[67], and transforming the structure into a single crystal by abnormal grain growth[68,69]. The final yield of the 3D printed NEMS devices is above 75%.

**Mechanical and vibrational properties**. The post-printing thermal process is the fundamental step to remove all the organic compounds and obtain resonators composed only of ceramic material with high-quality factor and frequency stability. Friction losses are responsible for the low performances of reported polymeric devices, like standard 3D printed resonators, limiting the quality factor below 100 (Figs. 1 and 3a for devices before

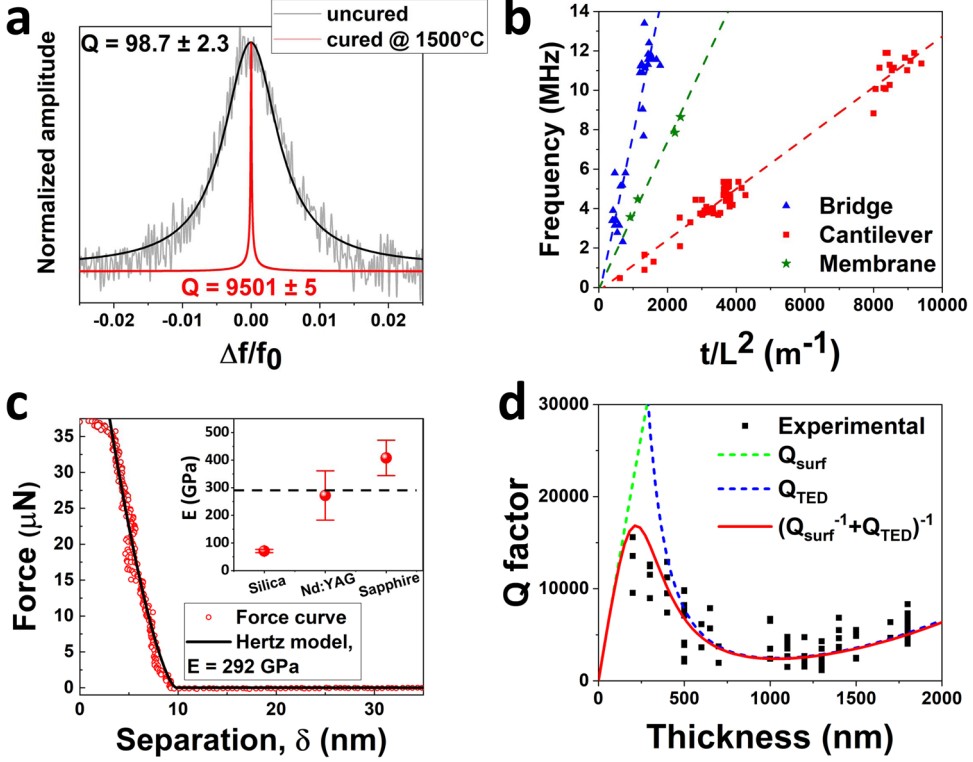

**Fig. 3 Resonance and quality factor analysis of 3D printed NEMS resonators. a** Amplitude spectra of cantilever device (pale lines) centered around the fundamental resonance mode before and after the thermal treatment. $Q$ values are extracted from Lorentzian fitting (thick lines). **b** Fundamental mode resonance frequency of the printed devices after thermal treatment as a function of the ratio thickness over square length $t/L^2$ (for membrane device $L$ is substitute with the radius). Dashed lines reported the plot of Eq. (1) for cantilevers, bridges, and membranes using material properties of Nd:YAG (Young's modulus and density). **c** Example of an AFM nanoindentation force curve as a function of the separation $\delta$ obtained on a membrane device. The line corresponds to the least-squares fit obtained by using the Hertz model with Young's modulus $E = 292\,GPa$, calculated by assuming a Poisson ratio equal to $\nu = 0.275$ for Nd:YAG. The inset reports the result obtained by averaging over 30 different points on the device along with the Young's modulus obtained on the nearby sapphire substrate and on a reference fused silica sample. **d** Measured $Q$ factor of 3D printed nanomechanical resonators as a function of device thickness. Dashed lines represent $Q$ contribution from surface loss (green line, Eq. (2)) and thermomechanical damping (blue line, Eq. (3)), the thick red line shows the resulting $Q$ factor $Q^{-1} = Q_{surf}^{-1} + Q_{TED}^{-1}$.

post-printing thermal process). In ceramic devices, the friction losses are reduced by three orders of magnitude[29], thus being not the limiting element on the quality factor. It was observed that the $Q$ of the printed resonators increases by two orders of magnitudes after the heat treatment (Fig. 3a). Further confirmation of the conversion into rigid materials comes from the analysis of the NEMS resonance frequencies after the thermal process. Fundamental resonance frequency $f_0$ of a mechanical resonator dominated by bending rigidity[11,30] is:

$$f_0 = A(E/\rho)^{1/2} t/L^2 \qquad (1)$$

where $E$ is Young's modulus, $\rho$ is the mass density, $t$ and $L$ are the thickness and length of the resonator as measured by SEM imaging (for circular membrane the length is substituted with the radius) and $A$ is a modal coefficient with value 1.028 for bridges, 0.162 for cantilevers and 0.469 for membrane. Figure 3b reports all the NEMS resonance frequencies as a function of $t/L^2$ ratio which well agree with the theoretical predictions (dash lines) obtained from Eq. (1) using the literature values for Nd:YAG of $E = 290\,GPa$ and $\rho = 4550\,kg/m^3$ [61]. Data confirm the absence of significant tensile stress and the resonators can be considered in a bending rigidity regime. The printed devices are completely converted into rigid structures with Young's modulus higher than silicon and comparable to silicon nitride one, as confirmed by independent analysis of stiffness from thermomechanical resonator motion and atomic force microscopy (AFM)

nanoindentation (see Supplementary Note 2). Both measurements technique confirm that the Young's modulus of the devices corresponds to that of Nd:YAG[61,70]. Figure 3c reports an example of nanoindentation force curve fitted to a Hertz model[71] with $E = 292\,GPa$. The inset shows the results obtained over 30 different points on the device. Results from the sapphire substrate and those obtained on a reference sample (fused silica) are reported as well, as a comparison.

The quality factor of the printed NEMS has been analyzed with three different equivalent methods, driving and measuring the resonator in its linear regime, measuring the thermomechanical motion of the resonators (i.e., Brownian motion), and evaluating the ring-down time of the device. In the first two methods, $Q$ is extracted from the Lorentzian fitting of the square of the amplitude motion signal. In the ring-down approach, $Q$ is computed from the energy dissipation of the damped resonator by fitting the exponential decay of the amplitude signal after stopping the actuation. Thermomechanical measurement is more reliable because independent of an external driving force, but since it is based on a very small resonator vibration induced by thermal force, it represents the noisier approach. Lorentzian fitting of driven resonator linewidth is very accurate while the device is in its linear regime and the $Q$ is not very high, up to a point where the width of the vibration peak is comparable to experimental set-up frequency resolution. The ring-down method instead is particularly used for very high $Q$s and long retention times. All the three methods applied to our bridges, cantilever and

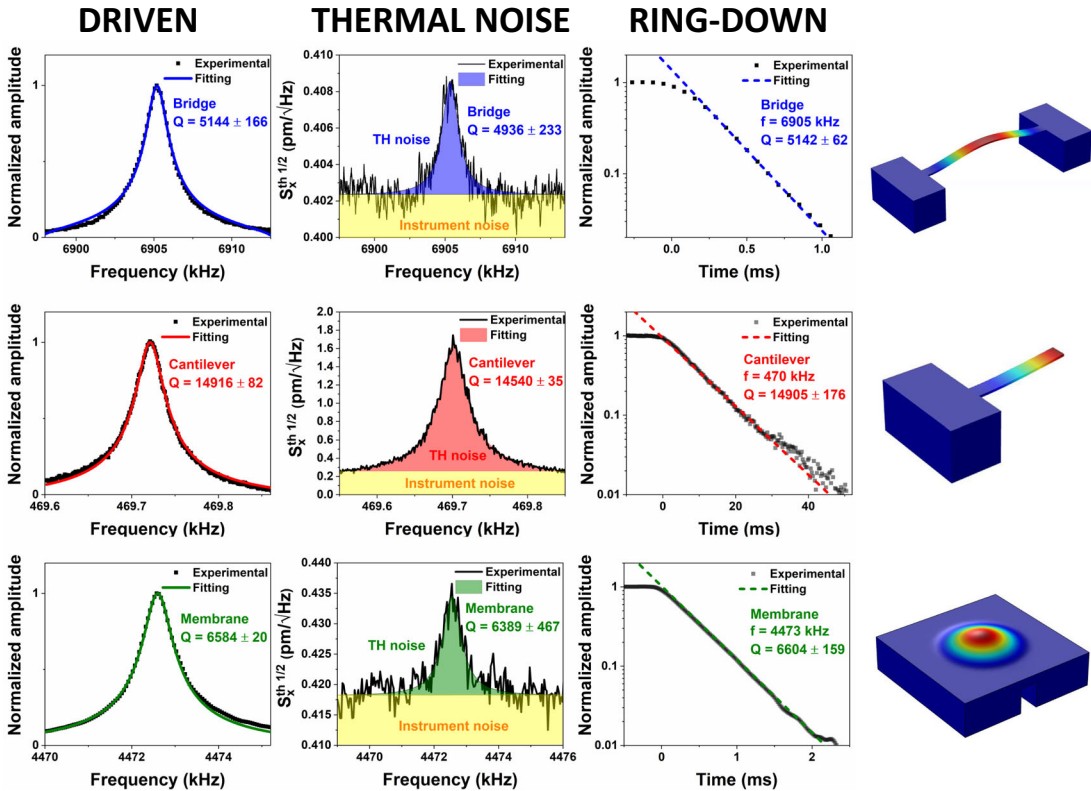

**Fig. 4 Quality factor analysis with different experimental approaches.** Experimental quality factor of bridge (upper panel), cantilever (central panel), and membrane (lower panel) devices extracted with three different approaches: driven the resonator to resonance, thermomechanical motion, and ring-down.

membrane resonators gave very consistent measurements within the experimental and fitting errors (Fig. 4). In a condition of high vacuum ($p \sim 10^{-7}$ mbar), 3D printed rigid nanomechanical resonators show quality factors from 1500 up to 15,000 (data reported in Fig. 3d), a range consistent with the quality factor of semiconductor unstressed NEMS (Fig. 1). We observe a strong dependence of Q with the thickness of the resonators. Higher Q are observed for thinner devices (around 200–400 nm) with a decrease of $t$ up to 1000 nm. For higher thickness Q shows a monotonic increase. Resonator damping is not limited by friction losses because of the conversion from soft to rigid materials, as described before, neither by radiation loss at the clamping which gives a contribution for all thickness, $Q_{\text{clamp}} > 10^5$ (details in Supplementary Note 3). The quality factor of our devices is dominated by two factors, surface friction ($Q_{\text{surf}}$) and thermoelastic damping ($Q_{\text{TED}}$). Surface loss caused by surface roughness, impurity, and adsorbates is a fundamental damping source in nanometric thick resonators because of the high surface to volume ratio. For wide resonators, $Q_{\text{surf}}$ has a linear dependence with resonator thickness $t$ as:

$$Q_{\text{surf}} = \frac{E}{6h_s E'} t \qquad (2)$$

where $h_s$ is the surface layer thickness and $E'$ the complex Young's modulus[30,72]. Thermoelastic damping is generated by the temperature gradient across the resonator thickness induced by the strain due to flexural vibration. $Q_{\text{TED}}$ dependence over the resonator thickness is more complex and can be described by the Zener model as

$$Q_{\text{surf}} = \left( \Delta_E \frac{\omega \tau_E}{1 + (\omega \tau_E)^2} \right)^{-1} \qquad (3)$$

with $\omega$ the resonator eigenfrequency, $\Delta_E = E\alpha^2 T/C_p$ the relaxation

strength and $\tau_E = t^2/\pi^2\chi$ the relaxation time ($\alpha$, $C_p$, and $\chi$ are the material thermal expansion coefficient, heat capacity, and thermal diffusivity, respectively)[73]. The resulting quality factor $Q(t)^{-1} = Q_{\text{surf}}^{-1} + Q_{\text{TED}}^{-1}$ plotted as a function of the thickness (red line in Fig. 3d) well describes the experimental Q of our devices. Below 600 nm, damping is governed by a combination of the two factors, while for higher thicknesses the losses are only dependent on thermoelastic damping.

**Frequency stability and mass sensitivity**. Frequency stability of a resonator, predicted for NEMS by Roukes et al.[74,75] as $\langle \delta f/f_0 \rangle \sim (1/2Q) 10^{-DR/20}$, is not only dependent on Q, but also on the dynamic range $DR$, the power level associated with the ratio between the maximum linear driven amplitude and the noise amplitude. Amplitude vibration of printed NEMS has been tested under different piezodisk voltage actuation to evaluate the linear range up to the onset of nonlinearity. Above the maximum linear driving amplitude, the resonators show typical shark-fin resonance lineshape due to geometrical nonlinearity (i.e., Duffing nonlinearity) with amplitude-dependent resonance frequency (Fig. 5a) and lineshape dependence over the sweeping frequency direction (Fig. 5b). The ratio between the thermomechanical noise signal and the maximum linear driven signal for the cantilever device yields to a large dynamic range $DR \sim 76$ dB (Fig. 5c) in line with other reported NEMS resonator[76]. Theoretical frequency stability of around $10^{-8}$ is expected from the above formula. The experimental frequency stability has been measured for all three device families with open-loop Allan deviation for integration time between 0.5 ms and 30 s (Fig. 6a). Minimum of Allan deviation is registered in the range 0.1–1 s with values of $0.7 \times 10^{-9}$, $1.2 \times 10^{-8}$, and $4 \times 10^{-8}$ for cantilever (as predicted above for single-clamped structures), bridge and membrane device, respectively. With these frequency stability

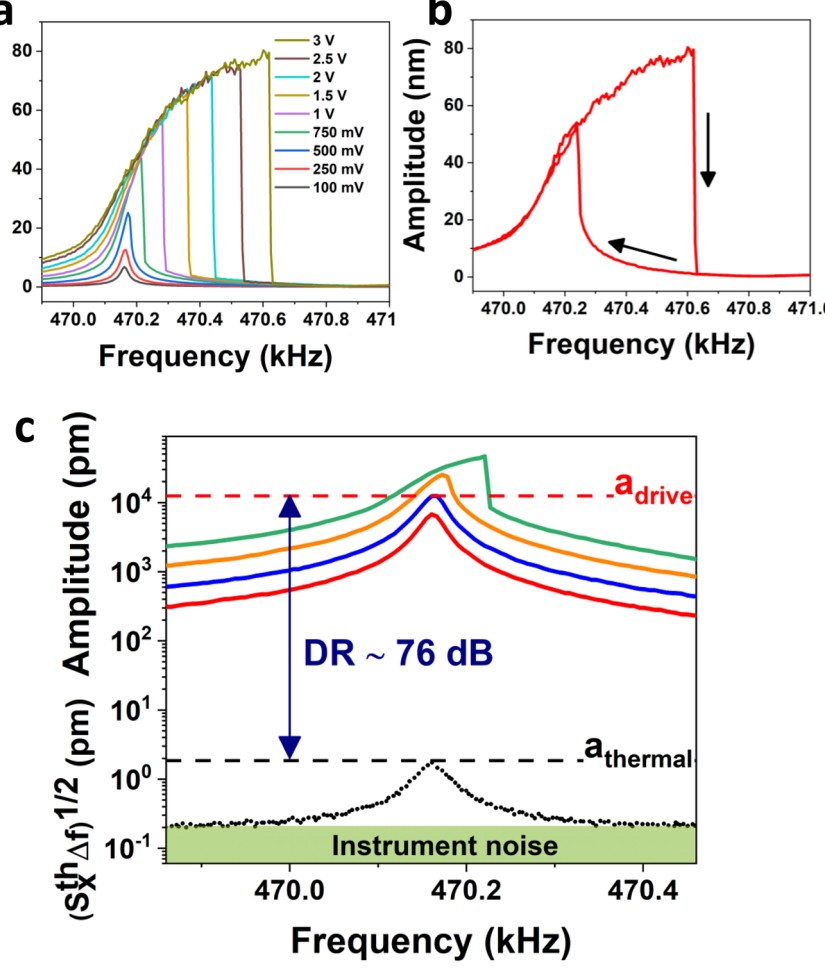

**Fig. 5 Dynamic range of 3D printed devices. a** Amplitude spectra of cantilever device for different piezodisk driving voltages. Above 750 mV, resonance curves show Duffing lineshape due to geometrical nonlinearity. **b** Nonlinear resonance frequency dependence over frequency sweep direction (indicated by the arrows). **c** Dynamic range of 3D printed cantilever between the amplitude of thermal noise spectrum and maximum linear driven signal.

values, the theoretical mass sensitivity of the 3D printed rigid NEMS is in the attogram range with a minimum of 0.45 ag for cantilever devices with 200-nm thickness. To compare the frequency stability and mass sensitivity with other devices in literature, we integrate the performances of our best resonators in a literature review plot presented by Sansa et al. (with the addition of some more recent works, Fig. 6b)[45]. Our devices have very good performances in line with the top-down NEMS family and general trend over device mass ($\propto m^{-1/2}$), confirming that approach is a valid alternative to silicon-based technology, but while using a much simpler and flexible fabrication technique. Demonstration of mass sensing capability of the 3D printed resonator is shown in Fig. 6c and d. A test mass (silica sphere with 0.5-μm diameter, details in the "Methods" section) has been deposited close to the cantilever tip causing a frequency shift of resonance peak of around 2 kHz. From the resonance frequency shift, a value of 116.6 fg of adsorbed mass can be computed, which is in line with the mass of a single silica bead of 124 fg (estimated from data provided by the distributor).

In addition to ultralow mass detection, the high $Q$ and lower mass make our resonators a very good candidate for highly sensitive force sensors. Force sensitivity is ultimately limited by thermal fluctuation to a value of 3.7 fN/√Hz for a cantilever

device computed as:

$$dF = \sqrt{4k_{\text{eff}} \frac{k_b T}{2\pi f Q}} \qquad (4)$$

where $k_{\text{eff}}$ represents the effective spring constant or stiffness extracted from the thermal noise spectrum of Figs. 4 and 5c, which represents a high sensitivity for room temperature nanomechanical sensors (details on the computation of effective stiffness in Supplementary Note 2). Higher sensitivity could be reached with strain-engineered phononic crystal devices, which on the counterpart are much more complicated to fabricate and have larger overall dimensions due to millimetric damping dilution structures[58,59].

**Discussion**

We have demonstrated the fabrication of NEMS resonators by 3D printing by two-photon polymerization technique. Soft printed structures are converted into rigid ceramic devices, by removing all the organic materials, thus obtaining nanomechanical resonators with performances in line with standard NEMS devices, which are currently fabricated with silicon-based technology. Our devices present a breakthrough alternative solution for ultralow mass sensing and force detection since they can be fabricated with

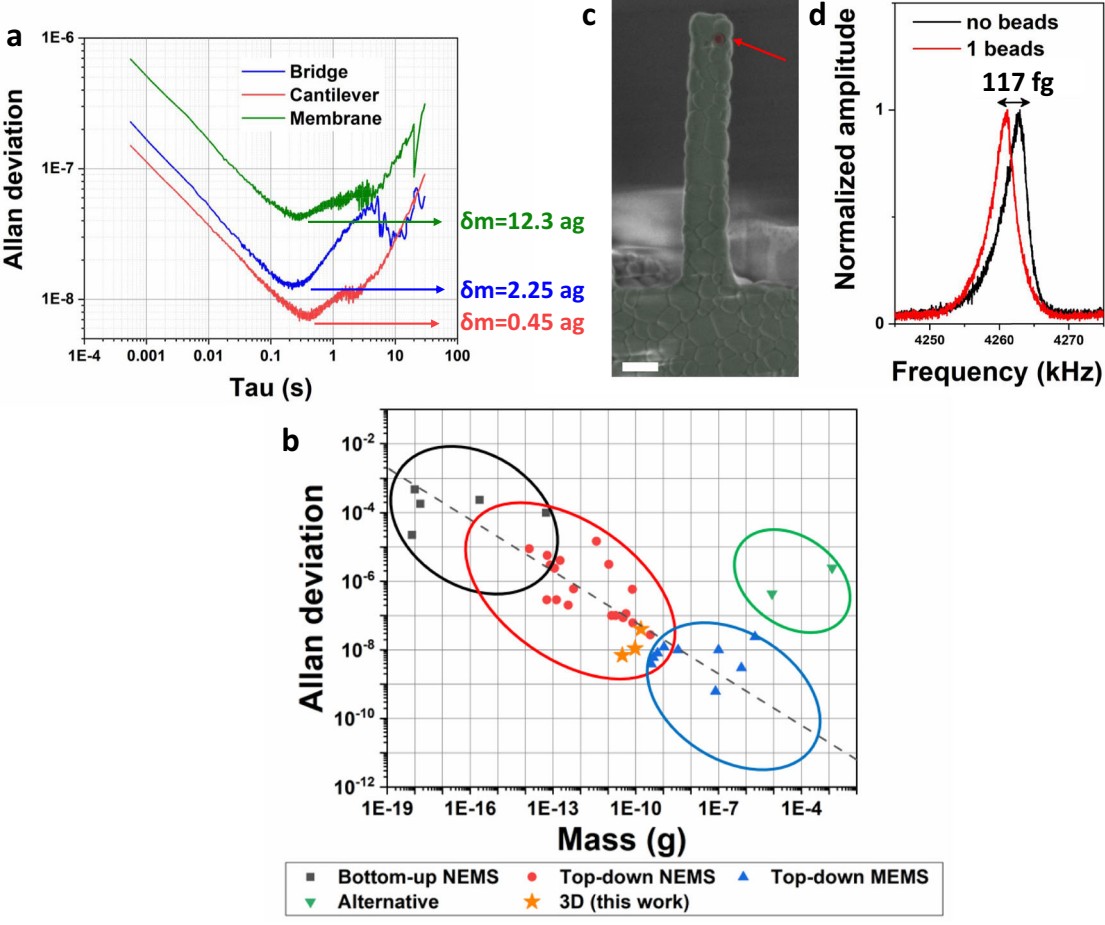

**Fig. 6 Frequency stability of 3D printed devices and mass sensing. a** Open-loop Allan deviation for the three types of resonators. Minimum detectable mass for each device is computed as $\delta m = 2m\delta f/f_0$. **b** Frequency stability of mechanical resonators from literature as function of device mass and fabrication approach. The graph is an integration with our devices (stars) and more recent literature works of the analysis presented by Sansa et al.[45]. Dashed line represents the general trend $10^{-12.2}\,m^{-1/2}$ reported in ref. [45]. References of literature work are reported in Supplementary Figs. S10, S11, and Supplementary Note 3). **c** SEM image in false colors of a 3D printed cantilever after silica bead adsorption. Silica bead is evidenced by red color and red arrow. Scale bar corresponds to 2 μm. **d** Resonance frequency peaks of a cantilever device before and after mass addition. The frequency shift corresponds to a mass addition of around 117 fg.

a simple, and versatile method, that can be utilized for fabrication of small numbers of NEMS devices or quick evaluation of prototypes before moving into large scale serial production. Although the process includes a heating step that may challenge integration with other devices, it could be possible in some applications to have the printing of resonator devices as the first process step and proceed with the additional technological steps after the heating process, or by moving the final crystalline device by a pick and place process, as shown in Supplementary Fig. S12. Moreover, the fields of NEMS and nanomechanics will gain much more attention from research groups that today are limited in access to clean room technology. In addition, our rapid prototyping method allows the possibility to create printed material with intrinsic functionalities by tailoring the starting precursor solution. Therefore, this uniqueness of the fabrication process can bring to the realization of new types of nanomechanical multiphysical devices. For example, Nd:YAG material presented in this work is an optical emitter at 1064 nm and can be the base for the fabrication of an integrated optomechanical device.

## Methods
### Samples fabrication
*Materials*. Yttrium chloride hexahydrate ($YCl_3 \cdot 6H_2O$) and neodymium chloride hexahydrate ($NdCl_3 \cdot 6H_2O$) were acquired from Strem Chemicals (USA).

Aluminum chloride hexahydrate ($AlCl_3 \cdot 6H_2O$) was purchased from Alpha Aesar (USA). Novec 7100, propylene oxide (PO), ethylene glycol, and the photoinitiator (PI) 4,4-Bis(diethylamino)benzophenone (BDAB) were acquired from Merck (Sigma-Aldrich, Israel). The photoinitiator 2-Benzyl-2-dimethylamino-1-(4-morpholinophenyl)-butanone-1 (IRG 369) were kindly given by IGM resins (Netherlands). Sapphire substrates were purchases from Gavish company (Israel).

*Polymerizable ceramic ink*. Precursor ink preparation and printing process were performed with a protocol already described in ref. [31]. Precursor Nd:YAG ink was prepared by dissolving 11.2 wt% of $YCl_3 \cdot 6H_2O$ in 41.7 wt% of triple distilled water (TDW) and ethylene glycol (65% ethylene glycol in TDW) at room temperature. After 5 min of stirring, 13.7 wt% of $AlCl_3 \cdot 6H_2O$ was added, followed by the addition of 0.7 wt% $NdCl_3 \cdot 6H_2O$. After further 30 min of stirring 4.1 wt% of acrylic acid was added and stirring was continued. Then after other 30 min, 27.8 wt% of propylene oxide was added in two steps (half of the amount while stirring, and then the second half 1 min later). After an additional 40 min of stirring, 0.4 wt% of PI IRG 369, and 0.4 wt% PI BDAB were added, and the mixture was stirred up to the complete dissolving of the photoinitiator components.

*3D printing process*. The two-photon printings were performed with a Photonic Professional GT printer (Nanoscribe GmbH, Germany). Precursor ink was placed between a glass and a sapphire disk spaced 140 μm apart by a custom-made metal spacer (as can be seen in Fig. 2). The micron-size structures were printed with ×25 magnification lens and nano-size structures with ×63 magnification lens. After the printing step, the obtained structures attached to the sapphire base were washed by immersing in ethanol for 5 min and then in Novec 7100 for 1 min to remove uncured ink.

*Thermal curing profile*. After the printing step, the devices were heated in a tube furnace (Zhengzhou Kejia Furnace, China) under air environment at 200 °C for

2 h, then to 520 °C for 2 h, 620 °C for 5 h, and finally to 1500 °C for 5 h. All the heating steps were reached with a heating rate of 0.6 °C/min.

**Device characterization**. Device dimension measurements and energy dispersive X-ray (EDX) analysis are performed with a Zeiss MERLIN field emission scanning electron microscope.

AFM nanoindentation measurements were performed by using an AFM (Innova, Bruker) equipped with a diamond nanoindenting probe (DNISP-HS, Bruker) mounted on a stainless-steel cantilever with $k = 353$N/m. The tip apex radius of the tip was 40 nm, the Young's modulus and Poisson ratio of the tip were, respectively, $E_t = 1140$ GPa and $\nu = 0.2$. A sapphire sample was used for calibrating the cantilever sensitivity while fused silica was used as a reference for the Young's modulus determination. The force curves were analyzed by using the Hertz model.

The frequency response of the printed mechanical resonators is measured using a Laser Doppler Vibrometer (LDV MSA-500, Polytec Gmbh). The sapphire disk with the resonators is mounted with an adhesive tape on a piezoelectric disk used for actuation. The vibrational spectra are recorded actuating the piezodisk with a sinusoidal chirp signal generated by the LDV system, in the specific frequency range of interest and evaluating the device response with FFT techniques (fast Fourier transform). All the vibrational measurements are performed at room temperature and vacuum level of $2 \times 10^{-7}$ mbar in a chamber evacuated by a vacuum system composed of a membrane and a turbomolecular pumps (HiCube80 Eco, Pfeiffer).

The frequency fluctuations of the 3D printed devices are evaluated by means of the Allan deviation measured by analyzing the voltage signal extracted from LDV with a lock-in amplifier (UHFLI, Zurich instruments). The Allan deviation is evaluated in an open-loop configuration by measuring the response of the resonator actuated at a fixed driving frequency (the resonance frequency). The phase of the vibration signal is monitored and then transformed into frequency values from the phase response of the resonator, which is linear close to the resonance frequency. The frequency values are then used to compute the Allan deviation, $\sigma_a$ in the integration time $\tau$:

$$\sigma_a = \sqrt{\frac{1}{2(N_a - 1)} \sum_{i=2}^{N} \left( \frac{\bar{f}_i - \bar{f}_{i-1}}{f_0} \right)^2},$$

where $\bar{f}_i$ is the time average of the frequency measurement in the $i$th time interval of duration $\tau$, $N_a$ is the total number of time intervals, and $f_0$ is the mean resonance frequency over the duration of the measurement.

Ring-down and nonlinearity measurements are performed with the lock-in amplifier used both for the piezodisk actuation and for the analysis of the voltage signal extracted from LDV.

Mass sensing tests are performed by deposition of a droplet of silicon dioxide beads (56796, MERCK) dispersed in Ultra-Pure water dispensed from a DirectQ-3UV Merck-Millipore (Italy) and then dried on a hot-plate at 80 °C. Nominal diameter dimension and density were 500 nm and 1900 kg/m³, respectively.

## Data availability
The data that support the findings of this study are available from the corresponding authors upon reasonable request.

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

## Acknowledgements

This research was partly supported by Israel Ministry of Science and Technology and the National Research Foundation, Prime Minister's Office, Singapore under its Campus of Research Excellence and Technological Enterprise (CREATE) program, the Ministero dell'Istruzione, dell'Università e della Ricerca (MIUR) through PRIN2017 - Prot.20172TZHYX grant and the European Commission through EU H2020 FET Open "Boheme" Grant No. 863179.

## Author contributions

S.S. and I.C. designed the experiment. I.C. and S.M. carried out the design and fabrication of the devices. S.S. carried out the vibrational measurements. M.T. carried out the indentation measurements and analysis. S.S., C.F.P., and C.R. processed and analyzed the vibrational data. S.S., I.C., and S.M. wrote the manuscript. All authors discussed the results and commented on the manuscript.

## Competing interests
The authors declare no competing interests.
