## [Peer Review File · Nature Communications]

Reviewers' Comments:

Reviewer #1:

Remarks to the Author:

The manuscript "Reaching silicon-based NEMS performances with 3D printed nanomechanical resonators" describes the mechanical properties of Nd:YAG suspended microstructures defined by photopolymerization. The authors have fabricated several tens of devices (cantilevers, bridge and membranes) and show that the performance is in line with NEMS devices made with semiconductor technologies. The characterization of the resonant behavior is extensive and according to the usual analysis in the field.

This reviewer agrees with the authors that the manuscript is of interest: the integration of functional in nanoelectromechanical devices would allow addressing application opportunities in various fields.

However, this reviewer believes that the manuscript should be improved in several aspects to show that this fabrication method can be adopted by the NEMS community and further developed to address significant applications. At the very least, the following aspects should be discussed in a revised version:

1) Dimensional control. The manuscript reports devices with dimensions in the range of microns and hundreds of nanometers. However, it is not mentioned in the manuscript with what precision dimensions are obtained. How far are the final dimensions from the target / designed dimensions? What is the dispersion in the final dimensions? The SEM images in Figure 2 show quite irregular and distorted structures. Can it be improved? SEM images reveal the polycrystalline structure of the resulting material, with a granular appearance. Can synthesis material be optimized to reduce grain size? Probably, this would allow to obtain better defined structures. In addition, the authors should explain how they measure the thickness of the devices at the end of manufacture.

2) Mechanical properties of the material. The agreement of the resonance frequency with the theoretical dependence is striking. However, an independent measurement of Young's modulus and density of the final material is still necessary to fully characterize the mechanical properties of the structures. Furthermore, it would be interesting to confirm whether the stiffness of the structures (elastic constant) is as expected. For example, from indentation experiments or measurement of thermal noise spectra.

3) In figure 1 and in several figures of the supplementary information the authors only use some of the devices in comparison with other mechanical resonators in the literature, with the comment "Some of our best devices are reported as star points". Authors should clarify how representative these devices are of all manufactured devices and what is the meaning of "best" as a criterion for selecting these devices.

4) The authors state that their method is simple and suitable for prototyping. This reviewer suggests adding a discussion in the manuscript to compare with other possible methods of synthesis and patterning of Nd:YAG and what are advantages and disadvantages, and if the method they propose can be expanded in some way to go beyond the laboratory scale. In addition, the fabrication method requires annealing at 1500 °C for 5 hours, which can limit its integration and hybridization with other materials.

5) The final statements of the manuscript (last paragraph beginning "in summary") indicates applications in integrated optomechanics, magnetism and quantum technologies. The authors should sustain these claims with some clear and specific arguments, otherwise it seems too speculative.

Reviewer #2:

Remarks to the Author:

The manuscript by Stassi et al describes the use of a ceramic material that can be additively

manufactured at the microscale for use in high quality factor microscale resonators. The process for making these resonators is a two photon polymerization direct laser writing followed by annealing to remove the organic components. The depth with which they present the background material, specifically Figure 1 and the associated SI figures, is truly commendable and does an excellent job putting their work into context. That said, this focus on background does highlight how the performance of the realized devices matches but does not exceed those produced using more conventional means. Further, the fabrication method itself is not novel as the same material was reported by some of the same authors in a prior work (their reference 15). Thus, the novelty here appears to be limited to the mechanical structures that are printed using this approach. In light of that, there is a lack of mechanical characterization beyond measurements of thermally-driven and actively driven frequency responses. Thus, taken together, it is not clear that this manuscript substantially advances the state of the art. My concerns are listed in more depth below.

(1) Describing the ceramic liquid ink as new seems incorrect in light of its earlier publication.

(2) Figure S4 presents EDX as evidence that carbon has been removed by the annealing process. To make this assertion, a spectra taken before the annealing process should also be provided as comparison.

(3) Mass sensing is consistently brought up as a major application of high sensitivity resonators. The authors also analyze the frequency response data to produce a predicted mass sensitivity. However, no measurements of mass are provided to evaluate these claims.

(4) Despite claims about the material properties (e.g. stiffness), the only measurements of mechanical properties come in the form of measurements of resonance frequencies, which measures the speed of sound in the material, not its stiffness. To make claims about the stiffness of the material, the spring constant of the devices should be measured. Also, the material properties of the manufactured material should be presented and compared with the tabulated values.

(5) Equation (4) leads to a prediction for the force resolution. In addition to the force resolution not being experimentally measured, it is reliant upon the effective spring constant. The process by which the spring constant is measured should be reported and these values should be presented in comparison with the resonance frequency values.

(6) Given that the performance of the resonators realized by this method are on par with the state of the art, it becomes very important whether the process for realizing them is simpler than the state of the art. It is claimed multiple times that this process is "much faster, simpler and flexible." While I agree with the flexibility inherent to additive manufacturing, it is hard to argue that strategies that can be parallelized (e.g. top down methods) are substantially slower than the present method unless a fair comparison is made and defended. Ultimately, this type of 3D printing – especially when considering the further constraints imposed by annealing – is also likely less precise than conventional MEMS processes that employ higher resolution lithography techniques such as electron beam lithography. Thus, I don't think the value proposition for this approach has been clearly or accurately articulated.

Comments on presentation:

-Nearly all the figures include text that is much too fine to read.

-Line 26 typo: "He 3D printed NEMS resonators..."

-Line 67: having an reference which is a superscript next to a symbol raised to a power is very confusing.

-Line 126 typo: "...increases in two orders of magnitude higher after..."

REVIEWER COMMENTS

REVIEWER #1 (REMARKS TO THE AUTHOR):

The manuscript “Reaching silicon-based NEMS performances with 3D printed nanomechanical resonators” describes the mechanical properties of Nd:YAG suspended microstructures defined by photopolymerization. The authors have fabricated several tens of devices (cantilevers, bridge and membranes) and show that the performance is in line with NEMS devices made with semiconductor technologies. The characterization of the resonant behavior is extensive and according to the usual analysis in the field.

This reviewer agrees with the authors that the manuscript is of interest: the integration of functional in nanoelectromechanical devices would allow addressing application opportunities in various fields.

However, this reviewer believes that the manuscript should be improved in several aspects to show that this fabrication method can be adopted by the NEMS community and further developed to address significant applications. At the very least, the following aspects should be discussed in a revised version:

QUESTION 1 *1) Dimensional control. The manuscript reports devices with dimensions in the range of microns and hundreds of nanometers. However, it is not mentioned in the manuscript with what precision dimensions are obtained. How far are the final dimensions from the target / designed dimensions? What is the dispersion in the final dimensions? The SEM images in Figure 2 show quite irregular and distorted structures. Can it be improved?. SEM images reveal the polycrystalline structure of the resulting material, with a granular appearance. Can synthesis material be optimized to reduce grain size? Probably, this would allow to obtain better defined structures. In addition, the authors should explain how they measure the thickness of the devices at the end of manufacture.*

ANSWER 1

We want to thank the reviewer for the questions.

In the revised manuscript, we addressed the following points:

Precision of dimensions

How far are the final dimensions from the target / designed dimensions?

What is the dispersion in the final dimensions?

Can deformation be improved?

Grain size

Measuring the thickness

- **Since all these comments are related, we revised the text to address the comments in the following text (Page 5):**

“The dimensional control depends mainly on the printing parameters and the shrinkage of the printed object during the thermal process after printing. The TPP process enables printing objects having features as small as 100 nm⁶². In our study, we start from a solution, obtain a hybrid object, followed by conversion of the hybrid structure (organic-inorganic) into an

inorganic, dense crystalline structure. These processes lead to a significant shrinkage, and therefore it is theoretically possible to go down to features in the range of tens of nanometers. After the thermal post-printing process, the printed resonators are composed of only inorganic polycrystalline Nd:YAG (as reported by EDX spectrum before and after thermal step, as shown in Supplementary Fig.S4) without any organic materials. As a result of solvent evaporation, burning of the organic material, and crystallization to the dense crystal structure, the material sintering is accompanied by a dimensions reduction. To compare the actual dimensions with the computer design file, we calculated the ratio between the measured dimensions of printed structures after the thermal treatment process and the theoretical ones (used for the design). The size measurements were made by SEM imaging, and the analysis was computed over more than 200 resonators. The Gaussian fit reports a mean value of 68.7 % of device isotopically shrinkage with a standard deviation of 5.3% (Supplementary Figure S5). Although the size reduction can help to achieve very small features, it could result in deformation of the final device geometries, especially for the circular membrane which is the most complicated to fabricate due to stress-induced during the thermal process (image of a device broken by thermal stress in Supplementary Figure S6). However, as it was presented in other publications, the deformation can be suppressed by printing the structures on guiding lines or domes^{63,64}. Furthermore, after the thermal post-printing, the surface becomes rough due to the crystallization of the structure (as seen in Supplementary Figure S6). To achieve a smoother surface, it is theoretically possible to gain smaller size grains by changing the heating conditions^{65,66}, selectively etch the YAG crystals with hot phosphoric acid⁶⁷, and transforming the structure into a single crystal by abnormal grain growth^{68,69}. The final yield of the 3D printed NEMS devices is above 75%.”

- **In addition, we clarify that the thickness was measured by SEM imaging of features of the objects. The following sentence is modified in the revised manuscript (Page 6): “...where E is Young’s modulus, ρ is the mass density, t and L are the thickness and length of the resonator as measured by SEM imaging” and the following sentence is added in the revised manuscript in the Method section (Page 15): “Device dimension measurements and Energy Dispersive X-ray (EDX) analysis are performed with a Zeiss MERLIN field emission scanning electron microscope.”**

QUESTION 2 2) Mechanical properties of the material. *The agreement of the resonance frequency with the theoretical dependence is striking. However, an independent measurement of Young’s modulus and density of the final material is still necessary to fully characterize the mechanical properties of the structures. Furthermore, it would be interesting to confirm whether the stiffness of the structures (elastic constant) is as expected. For example, from indentation experiments or measurement of thermal noise spectra.*

ANSWER 2

We agree with the review that a comparison of the mechanical properties of the material obtained by resonance frequency analysis with the values measured with other techniques would strengthen our work. Therefore, we follow both methods suggested by the reviewer. We analyzed the elastic modulus of the printed structures both from thermal noise spectra and by AFM

nanoindentation. Both techniques confirmed that the Young's modulus of the 3D printed nanoresonators is in line with the tabulated value of 290 GPa from previous literature works. **A comment on these analyses and comparison has been added in the main text (page 7).** *"The printed devices are completely converted into rigid structures with Young's modulus higher than silicon and comparable to silicon nitride one, as confirmed by independent analysis of stiffness from thermomechanical resonator motion and Atomic Force Microscopy (AFM) nanoindentation (see Supplementary Note 2). Both measurements technique confirm that the Young's modulus of the devices corresponds to that of Nd:YAG^{61,70}. Figure 3c reports an example of nanoindentation force curve fitted to a Hertz model⁷¹ with $E=292$ GPa. The inset shows the results obtained over 30 different points on the device. Results from the sapphire substrate and those obtained on a reference sample (fused silica) are reported as well, as comparison."*

An image containing an AFM force curve and a summary of the whole AFM nanoindentation measurement results has been added in figure panel 3 as figure 3c.

An extensive description of the used methods and measurements has been added in the Supplementary Information (Supplementary Note 2).

QUESTION 3 3) *In figure 1 and in several figures of the supplementary information the authors only use some of the devices in comparison with other mechanical resonators in the literature, with the comment "Some of our best devices are reported as star points". Authors should clarify how representative these devices are of all manufactured devices and what is the meaning of "best" as a criterion for selecting these devices.*

ANSWER 3

We want to thank the reviewer for the note.

For this work we tested more than 200 devices and it is impossible to show all of them in figure 1, while we report them in Figure 3b and d. The choice as best device has been done as a compromise of high Q value and low device mass. In figure 1 we reported two devices with high Q factor value and low mass for each resonator structure, cantilever, bridge and membrane. In the revised manuscript, we modify the caption of Figure 1 with a comment related to our choice "Two of our devices with the best performances in terms of high Q factor value and low device mass for each printed resonator structure (cantilever, bridge and membrane) are reported as star points."

QUESTION 4 4) *The authors state that their method is simple and suitable for prototyping. This reviewer suggests adding a discussion in the manuscript to compare with other possible methods of synthesis and patterning of Nd:YAG and what are advantages and disadvantages, and if the method they propose can be expanded in some way to go beyond the laboratory scale. In addition, the fabrication method requires annealing at 1500 °C for 5 hours, which can limit its integration and hybridization with other materials.*

ANSWER 4 :

We want to thank the reviewer for this note.

- **To address this comment, we have added the following text with comparison to other lithography techniques (page 2):**

“Another alternative technique to fabricate nanoresonators is two photons printing (TPP) lithography²³⁻²⁵. Based on multiphoton absorption, the polymerization occurs only at the focal point of an ultrafast laser (780 nm), leading to selective submicron size voxel curing within a droplet, hence providing the ability to “write” sub-micrometric structures²⁶. In contrast to better resolutions single exposure photolithography techniques, such as photolithography and electron beam lithography, the TPP technique enables achieving complex 3D structures without the need for multi fabrication steps. Another advantage is the ability to integrate structures made of different materials in the same substrate, by changing the printing resin within the droplet^{27,28}. One main disadvantage of TPP technology that prevented its adaptation in the industry compared to the traditional 2D printing techniques, is the slow printing process and the difficulty in making production at an industrial scale. However, due to the unique structures that can be fabricated by the TPP technology the industrial interest in this field is growing²⁹.”

- **Furthermore, we have added a sentence in the summary section that discusses the integration problem due to the heating process (page 12):**

“Although the process includes a heating step that may challenge integration with other devices, it could be possible in some applications to have the printing of resonator devices as first process step and proceed with the additional technological steps after the heating process, or by moving the final crystalline device by a “pick and place process”, as shown in supplementary figure S12.”

QUESTION 5 5) *The final statements of the manuscript (last paragraph beginning “in summary”) indicates applications in integrated optomechanics, magnetism and quantum technologies. The authors should sustain these claims with some clear and specific arguments, otherwise it seems too speculative.*

ANSWER 5

We agree with the Reviewer that especially last sentence was too speculative. We thus differentiated better our perspectives: on one side the technological point of view (no clean room), on the other the multiphysical device. In the latter, we just claimed about possible optomechanical use of Nd:YAG.

We modified the final paragraph (page 13) as follow: *“In addition, our rapid prototyping method allows the possibility to create printed material with intrinsic functionalities by tailoring the starting precursor solution. Therefore, this uniqueness of the fabrication process can bring to the realization of new types of nanomechanical multiphysical devices. For example, Nd:YAG material presented in this work is an optical emitter at 1064 nm and can be the base for the fabrication of an integrated optomechanical device”*

REVIEWER #2 (REMARKS TO THE AUTHOR):

The manuscript by Stassi et al describes the use of a ceramic material that can be additively manufactured at the microscale for use in high quality factor microscale resonators. The process for making these resonators is a two photon polymerization direct laser writing followed by annealing to remove the organic components. The depth with which they present the background material, specifically Figure 1 and the associated SI figures, is truly commendable and does an excellent job putting their work into context. That said, this focus on background does highlight how the performance of the realized devices matches but does not exceed those produced using more conventional means. Further, the fabrication method itself is not novel as the same material was reported by some of the same authors in a prior work (their reference 15). Thus, the novelty here appears to be limited to the mechanical structures that are printed using this approach. In light of that, there is a lack of mechanical characterization beyond measurements of thermally-driven and actively driven frequency responses. Thus, taken together, it is not clear that this manuscript substantially advances the state of the art. My concerns are listed in more depth below.

QUESTION 1 (1) *Describing the ceramic liquid ink as new seems incorrect in light of its earlier publication.*

ANSWER 1

Thanks for the comment, we agree with the reviewer that the ink is not new, since it has been already reported in a previous publication by our group, as also cited in the article. The term “new” related to the ink was reported only once in the abstract paragraph but was a refusal of manuscript editing and we have now removed it. The innovation in this work is the application of this ink and its printing for the fabrication of the NEMS resonator. The novelty is related to the possibility of reaching the performances of silicon-based resonators with a completely alternative approach. As we deeply underline with the literature analysis presented in Figure 1 and Figure 5b, many alternative fabrication approaches were tested in the last years, but no one could reach our results in terms of quality factor and sensitivity because they were all limited by the intrinsic loss of the material used and by the smaller device dimension that could achieve.

However, to more emphasize this issue we have added in the abstract (page 1): *“The devices are printed by a two-photon polymerization technique to reach nanometric resolution with our recently developed liquid ink composed of metal salts and photopolymerizable groups¹⁵.”*

QUESTION 2 (2) *Figure S4 presents EDX as evidence that carbon has been removed by the annealing process. To make this assertion, a spectrum taken before the annealing process should also be provided as comparison.*

ANSWER 2

We agree with the reviewer that showing EDX spectra before and after thermal curing will help the analysis of the burning and annealing process of the printed structures.

We have added in Supplementary Figure S4 of the SI the EDX spectrum of a printed structure before thermal curing and modified the caption accordingly. As expected, this spectrum confirms the presence of carbon, chlorine and fluorine related to the precursor material and to washing steps which are then removed during the thermal curing.

We have added a comment regarding EDX analysis before and after thermal curing in the main text (page 5): *“After the thermal post-printing process, the printed resonators are composed of only inorganic polycrystalline Nd:YAG (as reported by EDX spectrum before and after thermal step, as shown in Supplementary Figure S4) without any organic materials.”*

QUESTION 3 (3) *Mass sensing is consistently brought up as a major application of high sensitivity resonators. The authors also analyze the frequency response data to produce a predicted mass sensitivity. However, no measurements of mass are provided to evaluate these claims.*

ANSWER 3

We agree with the reviewer that a mass measurement test will improve the quality of our work confirming the capability of our resonators to be used as a mass sensing device. Therefore, we performed an experiment by depositing silicon dioxide beads with 500 nm diameter on our sample. Then we evaluate the resonance frequency shift of a device having a bead deposited as close as possible to cantilever tip. From the analysis of resonance frequency shift and knowing device mass we were able to compute the mass value of the silica beads of around 117 fg, close to 124 fg estimated from dimensions and density provided by distributor. This experiment confirms the capability of our device to be used as gravimetric sensor.

This analysis has been inserted in the final part of the manuscript, after frequency stability measurements adding Figure 6c and d in figure panel 6 and adding the following comment in the text (page 11): *“Demonstration of mass sensing capability of 3D printed resonator is shown in Figure 6c and d. A test mass (silica sphere with 0.5 μm diameter, details in Methods section) has been deposited close to cantilever tip causing a frequency shift of resonance peak of around 2 kHz. From the resonance frequency shift, a value of 116.6 fg of adsorbed mass can be computed which is in line with the mass of a single silica bead of 124 fg (estimated from data provided by distributor).”*

Details of the experiment has been added in the Methods section (page 16): *“Mass sensing tests are performed by deposit a droplet of silicon dioxide beads (56796, MERCK) dispersed in Ultra-Pure water dispensed from a DirectQ-3UV Merck-Millipore (Italy) and then dried on a hot-plate at 80°C. Nominal diameter dimension and density were 500 nm and 1900 kg/m³, respectively.”*

Instead, experimental evaluation of the ultimate mass sensitivity (minimum mass variation that could be detected) that we computed from the device frequency stability is much more complex. It would require the capability to deposit a single very low mass of few attogram (like a Double-stranded DNA molecule of around 1000 base pairs or a very small virus) on a device and also to be able to detect it with electron microscopy technique for adsorption and position confirmation.

QUESTION 4 (4) *Despite claims about the material properties (e.g. stiffness), the only measurements of mechanical properties come in the form of measurements of resonance frequencies, which measures the speed of sound in the material, not its stiffness. To make claims about the stiffness of the material, the spring constant of the devices should be measured. Also, the material properties of the manufactured material should be presented and compared with the tabulated values.*

ANSWER 4

We thank the reviewer for this comment on the material properties evaluation which was also risen by the other reviewer. As request by the reviewers, we evaluated the stiffness of the device from the thermal noise spectra using the equipartition theorem. Then, using the stiffness value, we computed the Young's modulus of the resonator comparing the formula of the device resonance frequency from lumped-element model and from Eulero-Bernoulli beam theory. The obtained values (related to thermal noise measurements) are in line with the tabulated value of Nd:YAG Young's modulus of 290 GPa. This evaluation is reported in detail in the Supplementary Note 2. Moreover, in the first version of the manuscript the elastic properties of the converted printed material (Young's modulus) were evaluated by comparing the value of the experimental resonance frequency of the device (Figure 3b) obtained with a driven measurement with the theoretical prediction from equation 1 (Eulero-Bernoulli beam theory) confirming that the printed devices have a Young's modulus in line with the literature value of 290 GPa. In the fundamental theoretical resonance frequency f_0 of the resonator from the Eulero-Bernoulli beam theory

$$f_0 = A(E/\rho)^{1/2}t/L^2$$

the term E represents the Young's modulus (or elastic modulus). The speed of sound in the elastic beam resonator is of course depending on the elastic modulus of the material, which is the value we evaluate in this analysis.

An additional comparison of the Young's modulus obtained by resonance frequency analysis has been made with new measurements performed with AFM nanoindentation technique which confirm that the printed devices have elastic properties in line with literature values for Nd:YAG.

A comment on these analyses and comparison has been added in the main text (page 7). *"The printed devices are completely converted into rigid structures with Young's modulus higher than silicon and comparable to silicon nitride one, as confirmed by independent analysis of stiffness from thermomechanical resonator motion and Atomic Force Microscopy (AFM) nanoindentation (see Supplementary Note 2). Both measurements technique confirm that the Young's modulus of the devices corresponds to that of Nd:YAG^{61,70}. Figure 3c reports an example of nanoindentation force curve fitted to a Hertz model⁷¹ with E=292 GPa. The inset shows the results obtained over 30 different points on the device. Results from the sapphire substrate and those obtained on a reference sample (fused silica) are reported as well, as comparison."*

An image containing an AFM force curve and a summary of the whole AFM nanoindentation measurement results has been added in figure panel 3 as figure 3c.

An extensive description of the used methods and measurements has been added in the Supplementary Information (Supplementary Note 2).

QUESTION 5 (5) Equation (4) leads to a prediction for the force resolution. In addition to the force resolution not being experimentally measured, it is reliant upon the effective spring constant. The process by which the spring constant is measured should be reported and these values should be presented in comparison with the resonance frequency values.

ANSWER 5

The procedure of computation of the spring constant (or stiffness) of the resonator is the same of the previous point. The effective spring constant is extrapolated from the thermal noise spectra of the devices using the equipartition theorem, as now described in the Supplementary Note 2. We used the value extracted from the thermal noise spectrum of Figure 3c to evaluate the theoretical force sensitivity of the cantilever device. This value represents the ultimate limit on force sensitivity of the device which considers only the intrinsic performance of the resonator and not the limitation due to external noise, measurement set-up and external interferences. Ultimate force sensitivity is a figure of merit presented in several works on nanomechanical resonators and thus we report such computation also in our work (*Science* 315, 490-493 (2007), *Science* 360, 764-768 (2018), *Nature Nanotechnology* 12, 776-783 (2017)).

To better contextualize and describe how we obtained this figure of merit, we modified the sentence related to force sensitivity as (page 11): “Force sensitivity is ultimately limited by thermal fluctuation to a value of 3.7 fN/√Hz for a cantilever device computed as:

$$dF = \sqrt{4k_{eff} \frac{k_b T}{2\pi f Q}} \quad (4)$$

where k_{eff} represents the effective spring constant or stiffness extracted from the thermal noise spectrum of Fig.3c and 4.c, which represents a high sensitivity for room temperature nanomechanical sensors (details on the computation of effective stiffness in Supplementary Note 2).”

QUESTION 6 (6) Given that the performance of the resonators realized by this method are on par with the state of the art, it becomes very important whether the process for realizing them is simpler than the state of the art. It is claimed multiple times that this process is “much faster, simpler and flexible.” While I agree with the flexibility inherent to additive manufacturing, it is hard to argue that strategies that can be parallelized (e.g. top down methods) are substantially slower than the present method unless a fair comparison is made and defended. Ultimately, this type of 3D printing – especially when considering the further constraints imposed by annealing – is also likely less precise than conventional MEMS processes that employ higher resolution lithography techniques such as electron beam lithography. Thus, I don’t think the value proposition for this approach has been clearly or accurately articulated.

ANSWER 6

We thank the reviewer for this comment, and revised the manuscript as follows:

1. **To be more precise and correct we have changed in the introduction and on page 2** “time consuming” into “multi-step”.

In the summary section, we have removed the word faster and modified the relevant sentence as follows: “Our devices present a breakthrough alternative solution for ultralow mass sensing and force detection since they can be fabricated with a simple, and versatile

method, that can be utilized for fabrication of small numbers of NEMS devices or quick evaluation of prototypes before moving into large scale serial production .”

2. In addition, we have added a discussion about two-photons printing, and its advantages and drawbacks over other processes such as photolithography and e-beam (page 2):

“Another alternative technique to fabricate nanoresonators is two photons printing (TPP) lithography²³⁻²⁵. Based on multiphoton absorption, the polymerization occurs only at the focal point of an ultrafast laser (780 nm), leading to selective submicron size voxel curing within a droplet, hence providing the ability to “write” sub-micrometric structures²⁶. In contrast to better resolutions single exposure photolithography techniques, such as photolithography and electron beam lithography, the TPP technique enables achieving complex 3D structures without the need for multi fabrication steps. Another advantage is the ability to integrate structures made of different materials in the same substrate, by changing the printing resin within the droplet^{27,28}. One main disadvantage of TPP technology that prevented its adaptation in the industry compared to the traditional 2D printing techniques, is the slow printing process and the difficulty in making production at an industrial scale. However, due to the unique structures that can be fabricated by the TPP technology the industrial interest in this field is growing²⁹.”

3. We have added on page 4: “Our devices are fabricated by printing a precursor solution ink with 2PP technique, followed by an additional heating step to elevated temperatures to transform the structure from hybrid to rigid crystalline material.”

Regarding the constraints imposed by thermal annealing, we have added a sentence in the summary section about the integration problem due to the heating process: “Although the process includes a heating step that may challenge integration with other devices, it could be possible in some applications to have the printing of resonator devices as first process step and proceed with the additional technological steps after the heating process, or by moving the final crystalline device by a “pick and place process”, as shown in supplementary figure 10.”

QUESTION 7 *Comments on presentation:*

-Nearly all the figures include text that is much too fine to read.

-Line 26 typo: “He 3D printed NEMS resonators...”

-Line 67: having an reference which is a superscript next to a symbol raised to a power is very confusing.

-Line 126 typo: “...increases in two orders of magnitude higher after...”

ANSWER 7

We thank the reviewer for this comment that will improve the readability of the work. We have corrected all the typo and increased the text dimension in the figures.

We have also splitted previous figure 3 and 4 in new figures (figure to enlarge images and text dimensions).

Reviewers' Comments:

Reviewer #1:

Remarks to the Author:

The authors have improved substantially the manuscript and in my opinion it is now in a publishable form.

Reviewer #2:

Remarks to the Author:

The authors have carefully considered my comments and those of the other reviewer and this led them to add further experiments and analysis. My comments have been addressed and I now support publication.

There are minor issues that should be fixed prior to publication (e.g. capitalization issues such as Mpa vs MPa and "1 beads" in the inset of Figure 5), but these can be addressed via careful copy editing.

REVIEWER COMMENTS

REVIEWER #1 (REMARKS TO THE AUTHOR):

The authors have improved substantially the manuscript and in my opinion it is now in a publishable form.

We thank the Reviewer for the positive comments and for the recommendation of publication of the work.

REVIEWER #2 (REMARKS TO THE AUTHOR):

The authors have carefully considered my comments and those of the other reviewer and this led them to add further experiments and analysis. My comments have been addressed and I now support publication.

There are minor issues that should be fixed prior to publication (e.g. capitalization issues such as Mpa vs MPa and "1 beads" in the inset of Figure 5), but these can be addressed via careful copy editing.

We thank the Reviewer for the positive comments and for the recommendation of publication of the work.

We have revised the typos underlined by the Reviewer and others that we have found in the manuscript.